# Complementarity of PALM and SOFI for super-resolution live-cell imaging of focal adhesions

Hendrik Deschout[1,*], Tomas Lukes[1,2,*], Azat Sharipov[1], Daniel Szlag[1], Lely Feletti[1], Wim Vandenberg[3], Peter Dedecker[3], Johan Hofkens[3], Marcel Leutenegger[4], Theo Lasser[1] & Aleksandra Radenovic[1]

Live-cell imaging of focal adhesions requires a sufficiently high temporal resolution, which remains a challenge for super-resolution microscopy. Here we address this important issue by combining photoactivated localization microscopy (PALM) with super-resolution optical fluctuation imaging (SOFI). Using simulations and fixed-cell focal adhesion images, we investigate the complementarity between PALM and SOFI in terms of spatial and temporal resolution. This PALM-SOFI framework is used to image focal adhesions in living cells, while obtaining a temporal resolution below 10 s. We visualize the dynamics of focal adhesions, and reveal local mean velocities around 190 nm min$^{-1}$. The complementarity of PALM and SOFI is assessed in detail with a methodology that integrates a resolution and signal-to-noise metric. This PALM and SOFI concept provides an enlarged quantitative imaging framework, allowing unprecedented functional exploration of focal adhesions through the estimation of molecular parameters such as fluorophore densities and photoactivation or photoswitching kinetics.

[1] Laboratory of Nanoscale Biology & Laboratoire d'Optique Biomédicale, STI - IBI, Ecole Polytechnique Fédérale de Lausanne, Station 17, CH-1015 Lausanne, Switzerland. [2] Department of Radioelectronics, FEE, Czech Technical University in Prague, Technická 2, 166 27 Prague 6, Czech Republic. [3] Department of Chemistry, University of Leuven, Celestijnenlaan 200F, B-3001 Heverlee, Belgium. [4] Abteilung NanoBiophotonik, Max-Planck-Institut für biophysikalische Chemie, Am Fassberg 11, 37077 Göttingen, Germany. * These authors contributed equally to this work. Correspondence and requests for materials should be addressed to T.La (email: theo.lasser@epfl.ch) or to A.R. (email: aleksandra.radenovic@epfl.ch).

I t is essential for cells to adhere to the extracellular matrix for carrying out fundamental tasks such as migration, proliferation and differentiation[1]. For all these processes, focal adhesions are essential. Focal adhesions rely on a concerted action of dense assemblies of hundreds of proteins[2] forming thin micron-sized plaques close to the cell membrane[3]. These protein assemblies contain transmembrane receptors, such as integrins, binding to the extracellular matrix and recruiting other proteins inside the cytoplasm, such as talin and paxillin. This entails the formation of small structures with an extent in the order of 100 nm, which either disassemble after a few seconds, or mature into larger focal adhesions that remain stable typically for tens of minutes. This underlying maturation process requires an ongoing recruitment of additional proteins, such as vinculin or α-actinin, which may be linked to actin filaments. Overall, focal adhesions can thus be seen as the anchor points of the cell onto the extracellular matrix, mediating interactions with the actin cytoskeleton. Most focal adhesion proteins have been identified. However, the observation of the spatial organization and dynamics of focal adhesions remains challenging.

Single-molecule localization microscopy (SMLM), based on localizing sparse sets of activatable or switchable fluorescent molecules with a precision of tens of nanometres, is considered to be a method of choice for this endeavour[4]. In 2006, Betzig et al.[5] used photoactivated localization microscopy (PALM) to image submicron patterns of vinculin in a fixed cell. However, focal adhesions are dynamic entities demanding fast live-cell imaging. This has been further investigated by using PALM to image the dynamic behaviour of paxillin[6], but elucidating the full cell adhesion process remains a challenging task for SMLM.

As shown by Shroff et al.[6], SMLM trades temporal resolution for spatial super-resolution, since using less raw images for individual SMLM images means less available single-molecule localizations. Several thousand raw images offer high spatial information of focal adhesions, but only a limited first glimpse into their dynamic behaviour. These focal adhesions not only evolve over time, they can also undergo translational movement. The mean velocity of focal adhesions in stationary fibroblasts has been reported to be in the order of $100 \, nm \, min^{-1}$ (ref. 7). This translates into a temporal resolution well below 1 min to capture the fundamental dynamic behaviour while avoiding motion blur, which would otherwise spoil the anticipated spatial resolution[6]. Although temporal resolutions in the order of seconds are possible using PALM[8], the SMLM method most often reported to achieve such a temporal resolution is (direct) stochastic optical reconstruction microscopy ((d)STORM)[9,10]. However, delivery of (d)STORM dyes to intracellular targets remains difficult[11]. PALM is well suited for live-cell imaging of focal adhesions since it uses genetically expressed fluorescent proteins known for being well tolerated in living cells.

PALM holds promise for obtaining information about the spatial composition and organization of proteins in focal adhesions. Indeed, assuming that each fluorescent protein is localized only once, their numbers would directly result in a fluorophore density map. However, fluorescent proteins are known to 'blink', that is, they can reversibly switch on and off for several times after being activated[12]. Blinking therefore results in an overcounting error. Several methods have been developed to account for this error, for instance, by combining localizations that are clustered in space and time[13,14] or by applying pair correlation analysis[15]. Undercounting errors can appear as well, not only by merging localizations of separate fluorophores in high-density samples, but also due to incomplete maturation and limited detection efficiency[16].

To address the need for quantitative and time-lapse super-resolution imaging of focal adhesions, we enlarge the scope of SMLM by merging PALM with super-resolution optical fluctuation imaging (SOFI)[17] applied to the same raw image sequence. SOFI exploits the correlated response of neighbouring image pixels based on a spatiotemporal cumulant analysis of image sequences[18]. This technique tolerates a significant overlap of single-molecule images and relaxes the requirements on the activation or switching rates when compared with classical SMLM concepts. This allows one to use fluorescent molecules with a higher activation or switching rate[19], resulting in an improved temporal resolution[20]. However, there is a common belief that SOFI cannot attain the spatial resolution achievable by known SMLM methods. In addition, balanced SOFI (bSOFI) can be used to determine the fluorophore on-time ratio, offering an estimation of the molecular density and molecular switching or activation rates[21].

In this paper, we investigate the complementarity of PALM and SOFI for imaging focal adhesions. By applying them both to the same data set, we obtain a better insight in the true structure of focal adhesions and their molecular parameters. We enhance bSOFI and achieve a substantial increase in spatial resolution, comparable to PALM. We also present a methodology for evaluating the super-resolution image quality, integrating a resolution and a signal-to-noise (SNR) metric. We demonstrate our PALM-SOFI framework by imaging moving focal adhesions in a living cell.

## Results

**Widefield super-resolution metrics.** In Abbe's theory, microscopy imaging is conceived as low pass filtering with a cutoff frequency at $2NA/\lambda$ (with $\lambda$ the wavelength of light and NA the numerical aperture of the microscope objective). Abbe's analysis established the generally adopted resolution metric for classical microscopy as a pure instrument parameter independent of the object. SMLM goes beyond the 'diffraction barrier' by exploiting to its best the precise localization of single fluorophores. Therefore, the final 'SMLM-resolution' is the accumulated information of localized fluorescent markers and is *de facto* sample dependent.

In recent publications[22,23], the concept of an optical resolution criterion was revisited with an extension to super-resolution imaging. However, as stated by Demmerle et al.[22], 'resolution in single-molecule imaging is especially challenging'. There is a manifold of sample dependent and difficult to master parameters like labelling density, bleaching and the sample structure itself, which have a difficult to assess impact on resolution. In view of merging different imaging modalities like PALM and SOFI, the need for a general resolution and SNR metric became mandatory.

An important step towards a resolution metric is the Fourier ring correlation (FRC)[24,25]. Essential to this metric is the correlation of the Fourier transform of two SMLM images obtained from two stochastically independent halves of the original image sequence (Supplementary Note 1). An extension of the FRC procedure applies also to SOFI, which we used for an objective assessment of PALM and SOFI. We imaged fixed mouse embryonic fibroblasts (MEFs) expressing paxillin labelled with mEos2 or psCFP2 (see Methods), and calculated the FRC metric as a function of the number of frames, as shown in Fig. 1. To improve the spatial resolution of SOFI, we introduced a novel linearization procedure for bSOFI to achieve higher orders of the cumulant analysis (Supplementary Note 2).

Figures 1a,b show that the individual adhesion footprints are structured into a specific pattern. As the FRC calculation involves circular path summing in frequency space with a constant radius, the FRC metric is almost insensitive to variations of the spatial frequency content along different directions (Supplementary Figs 1 and 2). In Fig. 1a,b, such a difference can readily be

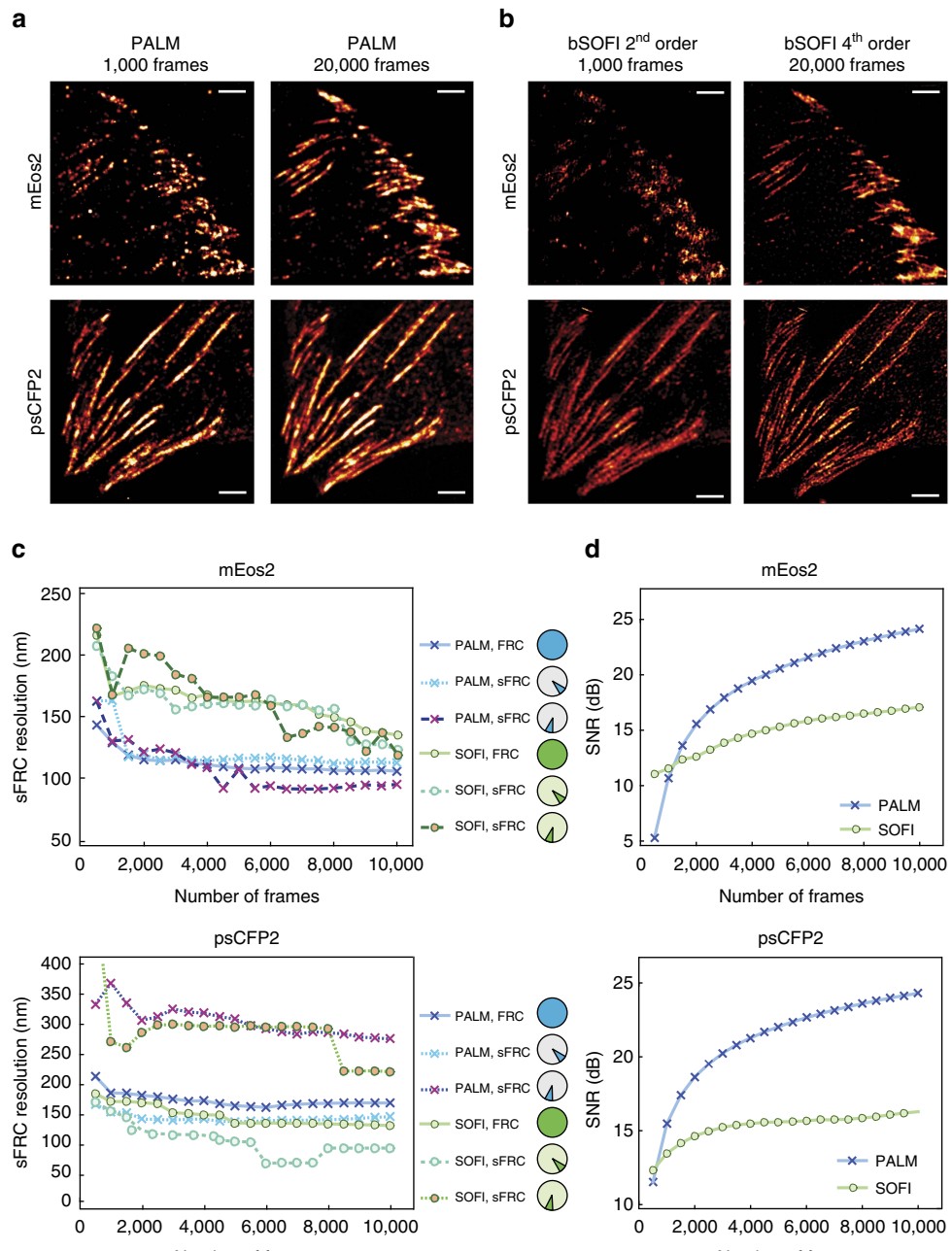

**Figure 1 | Objective image quality assessment integrating a resolution metric and a SNR metric applied to PALM and SOFI images.** (**a**) PALM images of fixed MEF cells expressing paxillin labelled with mEos2 or psCFP2, obtained from a full raw image sequence (20,000 frames) and the first 1,000 frames. The PALM images were rendered as probability maps (see Methods). (**b**) SOFI images obtained from the same raw image sequences as in **a**. (**c**) Resolution (sFRC) metric for SOFI and PALM as a function of the number of frames, obtained from subsequences of the same raw image sequences as in **a** and **b**. The circles indicate the sector used for the sFRC calculation, the sector with the lowest sFRC values provides the best description of the resolution. Note that the sFRC requires to split the number of frames in two halves to create two images. Therefore, 20,000 input frames allows one to calculate the sFRC corresponding to a super-resolved image reconstructed from 10,000 frames. (**d**) SNR metric for SOFI and PALM as a function of frames, obtained from subsequences of the same raw image sequences as in **a** and **b**. Scale bar, 2 μm.

noticed for the psCFP2 marked cell image, where focal adhesions and elongated structures indicative of paxillin organized along actin filaments[26] can be seen. We therefore implemented a sectorial FRC (sFRC) metric (Supplementary Note 1) as already suggested by Nieuwenhuizen *et al.*[25] This sFRC metric shows a more nuanced picture: the measured values are varying around the classic FRC for different sectors as shown in Fig. 1c and Supplementary Fig. 3, reflecting the orientation dependence of the

resolution metric. The resolution capabilities of the imaging technique are best described by the sector with the lowest sFRC value, indicating that a spatial resolution around 100 nm was obtained. Interestingly, the sFRC values indicate that SOFI resolves psCFP2-expressing cells better than PALM, while the opposite was observed for mEos2 labelling, despite the latter fluorescent protein being well known for its blinking properties. We attribute these results to a difference in activation rate and

emitter density, as indicated by the evolution of the number of localizations over time (Supplementary Fig. 4). The number of psCFP2 localizations is higher during the first several thousand frames, increasing the probability of overlapping psCFP2 images, which poses more difficulties for PALM than for SOFI.

Following these observations, we extended our PALM-SOFI framework to dual-colour imaging using both psCFP2 and mEos2. We imaged a fixed rat embryonic fibroblast (REF) expressing paxillin labelled with psCFP2 and integrin β3 labelled with mEos2 (see Methods). Calculating the sFRC metric in the two-colour channels for both SOFI and PALM shows that SOFI obtains the highest spatial resolution in the psCFP2 channel (that is, around 90 nm), while PALM gives the best resolution (that is, around 100 nm) in the mEos2 channel, in correspondence to our single colour results above. This suggests that an overlay of the SOFI (psCFP2) and PALM (mEos2) images results in an improved dual-colour image, as shown in Fig. 2.

Besides the image resolution, the image SNR should be characterized as well. We performed a pixel-wise SNR estimation based on a statistical approach known as jackknife resampling[27]. The jackknife method generates N data sets of N-1 camera frames, that is, each jackknife data set is obtained by 'cutting-out' just one single camera frame (Supplementary Note 3). The variance on the individual pixel values originating from each of these data sets is considered as an uncertainty measure, yielding an SNR value per pixel. This general approach applies to PALM as well as to SOFI and has been used as an objective comparison of SNR for our PALM and SOFI cell images, as shown in Fig. 1d. Except for a small number of frames (typically < 1,000), PALM outperforms SOFI in terms of SNR. This is to be expected because the PALM images are reconstructed from fitted data.

In summary, our methodology for assessing the image quality integrates an objective evaluation of the resolution and the SNR for super-resolved images.

**From spatial towards temporal resolution.** Achieving a high temporal resolution in SMLM is truly a challenge. Bleaching, activation or switching rates, camera frame rates and last but not least the minimum number of frames limit the achievable temporal resolution. As stated before, spatial super-resolution comes at an expense of temporal resolution. As we intend to image the dynamics of focal adhesions, we are in need of characterizing the difficult balance between lowering spatial super-resolution while enhancing temporal resolution. To objectively characterize the spatiotemporal resolution of both SOFI and PALM for a broad range of controlled conditions, we performed resolution measurements using simulated data. In an attempt to stay close to classical resolution measurement concepts, we designed a test target adopted from charts used for modulation transfer function (MTF) analysis. The MTF allows one to extract the cutoff frequency and the visibility as a function of spatial frequency of an imaging system and is used as a metric for characterizing optical imaging instruments[28]. Our MTF analysis provides a resolution standard for simulated data and a control for the sFRC resolution estimates in our high-density conditions.

Our test target consists of progressively smaller bars randomly filled with point emitters at an *a priori* given density, providing an object of stochastically activated single emitters (Fig. 3a; Supplementary Note 4). To approximate the conditions of focal adhesions in a cell, we tested two emitter densities (that is, 800 and 1,200 μm$^{-2}$). Our simulation takes into account the photophysics of mEos2 and psCFP2 and parameters of the microscope set-up (Supplementary Note 4). On the basis of this test target, we determined the visibility for PALM and SOFI beyond the cutoff frequency $f_c$ of classical widefield microscopy. From each simulated MTF, we extracted the cutoff frequency (Fig. 3b; Supplementary Note 4), resulting in a resolution measure related to the sFRC metric (Supplementary Fig. 5).

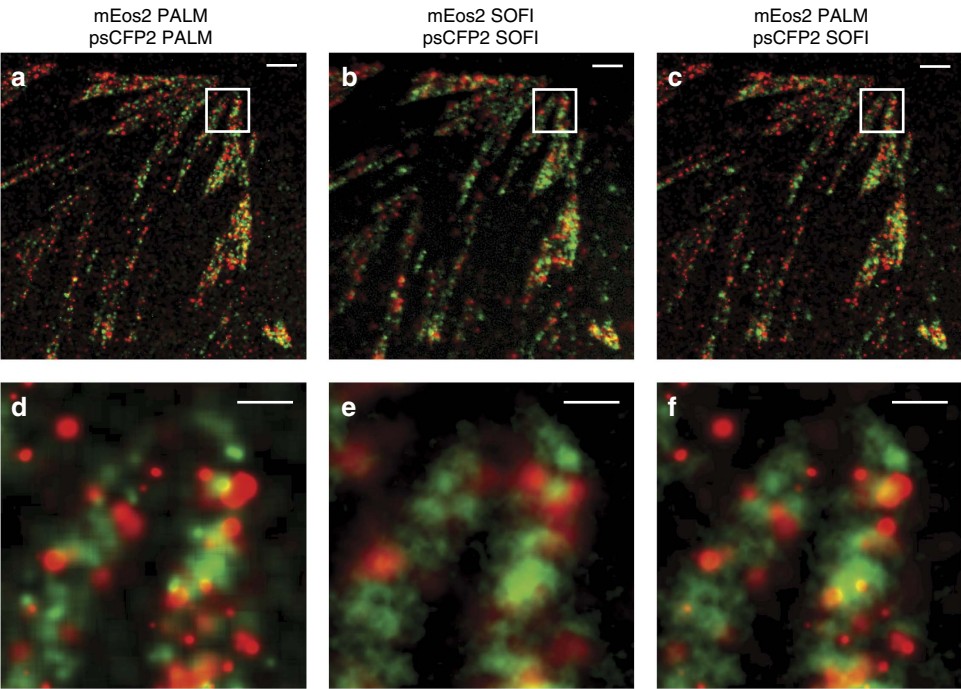

**Figure 2 | Dual-colour imaging with PALM and SOFI.** (**a–c**) Overlay of red and green images of a fixed REF cell expressing paxillin labelled with psCFP2 (green) and integrin β3 labelled with mEos2 (red) as obtained by (**a**) PALM in both colour channels, (**b**) SOFI in both colour channels, and (**c**) PALM in the red channel and SOFI in the green channel. (**d–f**) Corresponding zoom-ins for the images in **a–c**. The PALM images were rendered as probability maps (see Methods). Scale bar, 2 μm (**a–c**); 0.5 μm (**d–f**).

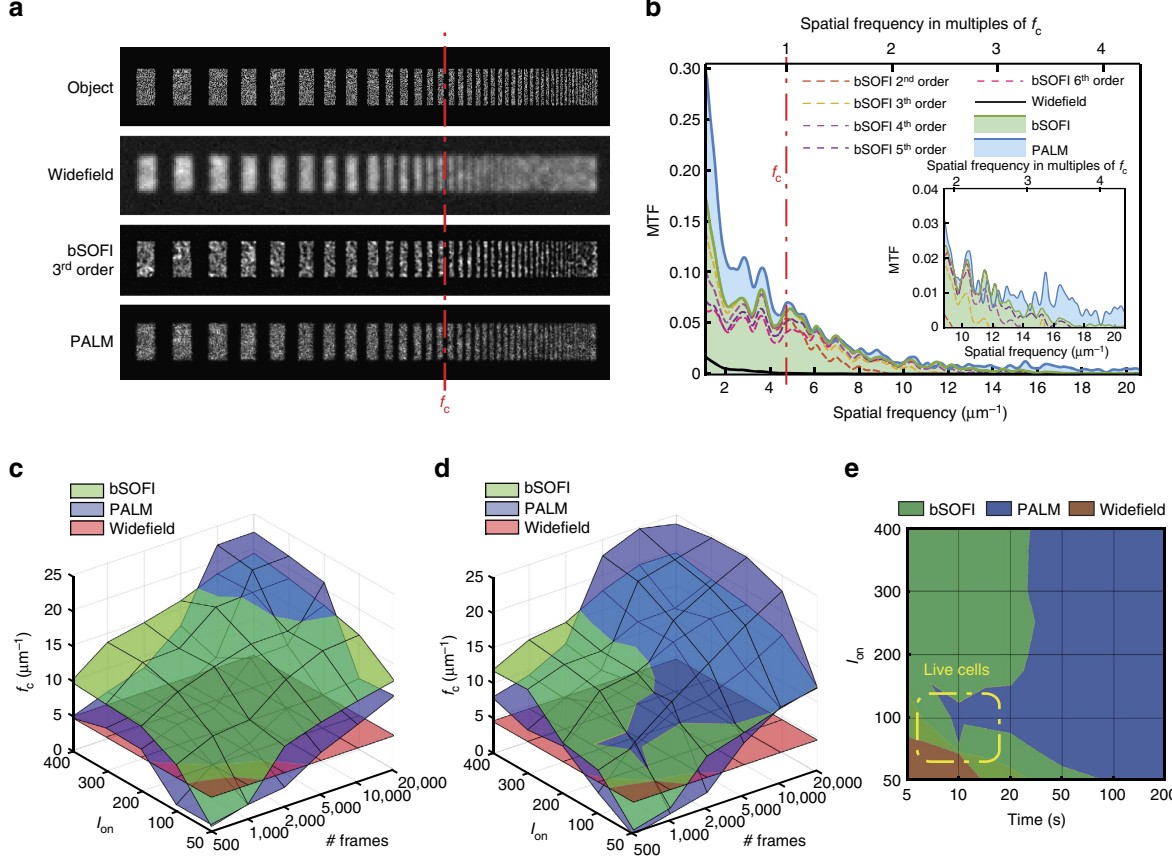

**Figure 3 | MTF analysis on simulated PALM and SOFI images.** (**a**) MTF test target consisting of single emitters randomly placed inside progressively thinner bars, together with resulting widefield, PALM and third order SOFI image ($I_{on} = 100$ photons and 20,000 frames). The red line indicates the cutoff frequency $f_c$ for widefield imaging. The PALM images were rendered as localization number histograms (see Methods) with a pixel size equal to the SOFI pixel size. (**b**) MTF calculated from the simulated SOFI and PALM images in **a**. (**c,d**) Cutoff frequencies for PALM and SOFI as a function of $I_{on}$ and the number of frames, with an emitter density of (**c**) 1,200 μm$^{-2}$ and (**d**) 800 μm$^{-2}$. (**e**) Two-dimensional projection of the chart in **d**. The timescale assumes a frame rate equal to 100 Hz, which corresponds to the frame rate used for our live-cell measurements.

Figure 3c,d shows the simulated cutoff frequency maps for PALM and SOFI based on the same test target, as a function of the number of frames and the number of photons per emitter per frame in an on-state (that is, $I_{on}$). Figure 3c corresponds to an emitter density of 1,200 μm$^{-2}$ and the psCFP2 case, whereas Fig. 3d corresponds to 800 μm$^{-2}$ and the mEos2 case. The number of frames ranges from 500 to 20,000. At 20,000 frames, all emitters are detected and the structure of the test pattern is fully described. SOFI shows a slowly growing spatial resolution (that is an increase of cutoff frequency $f_c$) with increasing $I_{on}$ and the number of frames. The PALM cutoff frequency grows faster, but only outperforms SOFI for a high number of frames (>10,000 for the higher density case and >5,000 for the lower density case). Note that SOFI requires at least 500 frames before 'super-resolution' can be achieved, while PALM needs even more frames (typically >1,000) and depends more strongly on the labelling density. For low frame numbers and low $I_{on}$, the number of localized emitters and the localization precision are too low for PALM to properly describe the test pattern, which results in low MTF values and a corresponding low resolution. Assuming a typical camera frame rate of 100 Hz, Fig. 3e shows the resolution sub-space where SOFI is dominant over PALM in terms of temporal/spatial resolution, and vice versa the sub-space where PALM outperforms SOFI. This indicates the parameter space where our PALM-SOFI imaging modality can be used for investigating the dynamics of focal adhesions as indicated in Fig. 3e.

**Live-cell imaging.** Imaging living cells requires a technique providing a sufficiently high temporal resolution and a compatibility with physiological conditions. Among the different SMLM methods, PALM meets the latter condition well, mainly due to genetically expressed fluorescent proteins acting as a label. However, the first condition is not perfectly met. PALM (like other SMLM techniques) makes the implicit assumption that the imaged structure stays stationary during the image acquisition, typically lasting for several minutes. Observing objects moving with a speed exceeding 10 nm min$^{-1}$ (that is the typical localization precision) is almost incompatible with this stationarity condition. Focal adhesions are known to move at rates of about 100 nm min$^{-1}$, as mentioned before. Observing focal adhesions therefore demands PALM imaging cycles far below 1 min, to avoid significant motion blur. The obvious way to increase the temporal resolution is to shorten the imaging cycle by acquiring less raw images. However, this entails a decrease in spatial resolution as less localizations are contributing. Many attempts have therefore been undertaken in SMLM to improve the temporal resolution, while maintaining a sufficient number of localizations[8,29–31].

SOFI offers a large untapped potential for imaging living cells. Just like PALM, SOFI can be used with genetically expressed fluorescent proteins. However, it is also assumed that the sample is stationary during the acquisition of the raw images. This again asks for a tradeoff between spatial and temporal resolution, although SOFI images can be reconstructed with less images than

required in PALM. When comparing SOFI and PALM, the latter technique is generally perceived as providing a higher spatial resolution. SOFI, on the other hand, is assumed to feature a higher temporal resolution, allowing faster imaging of moving structures, which has indeed been suggested by Geissbuehler *et al.*[20].

When attempting to increase both temporal and spatial resolution, a PALM-SOFI approach based on an identical raw image sequence appears as an interesting imaging modality. We imaged living MEF cells expressing paxillin labelled with mEos2 and post-processed the data by both PALM and SOFI algorithms, as shown in Fig. 4a and Supplementary Video 1. We obtained a temporal resolution of 10 s, while maintaining an average spatial resolution of 157 nm for SOFI, as determined by the sFRC metric (Supplementary Fig. 6). PALM at this temporal resolution resulted in an average spatial resolution of 145 nm. We determined the mean velocity of one of the focal adhesions, obtained from a kymograph-based analysis[32,33] (Fig. 4b,c;

Methods). PALM and SOFI show similar trends, indicating that the focal adhesion moved with a mean velocity of 190 nm min$^{-1}$. This mean velocity is in agreement with values reported and observed by others[7].

**Quantitative imaging.** Beyond qualitative imaging, SMLM methods such as PALM allow one to obtain quantitative molecular information, such as the number of localizations. This can be related to the number of fluorescent proteins. However, the relationship between both quantities is far from trivial, since most photoactivatable fluorescent proteins blink, that is they can reversibly go to a dark state. This may give rise to multiple localizations. Moreover, this blinking behaviour depends on the illumination intensity and the molecular environment of the fluorescent proteins. Simply counting the localizations usually results in an overestimation of the number of fluorescent proteins. Hence, several methods to correct this overcounting

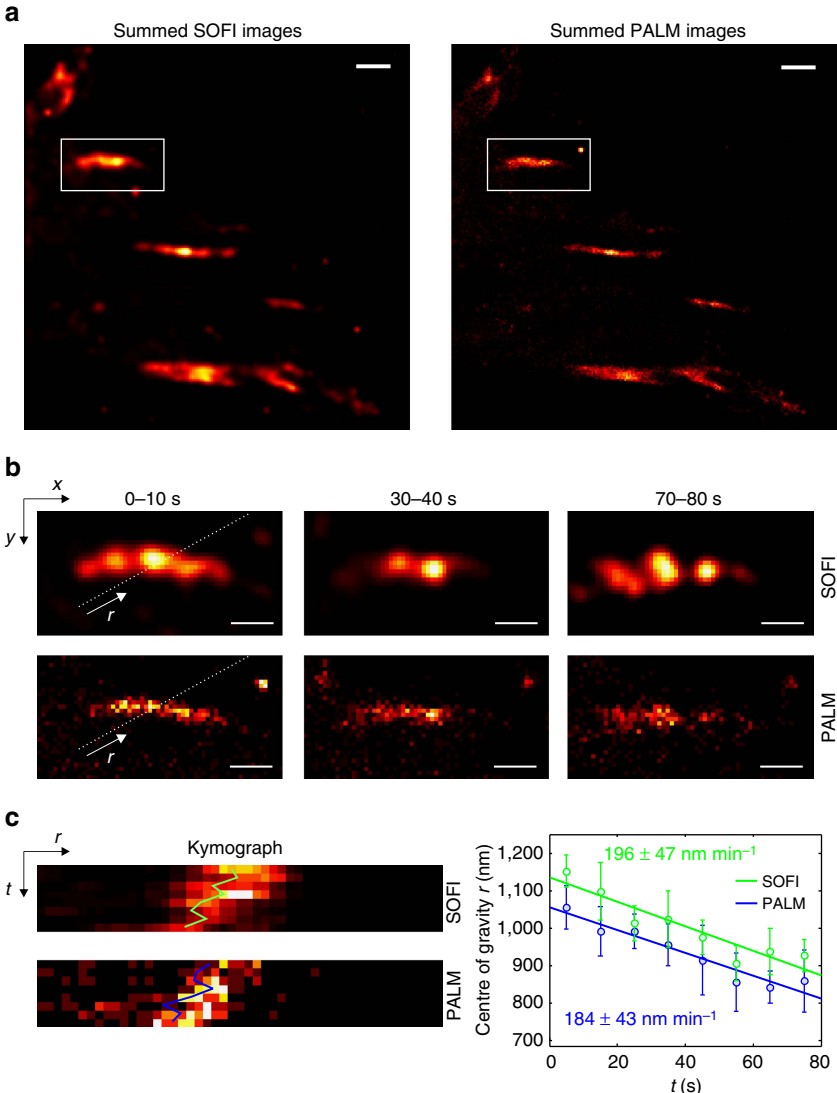

**Figure 4 | Live-cell imaging with PALM and SOFI.** (**a**) Sum of eight PALM and SOFI images of a living MEF cell expressing paxillin labelled with mEos2. Each image is reconstructed from 1,000 camera frames with 10 ms exposure time, resulting in a 10 s temporal resolution. The PALM images were rendered as localization number histograms (see Methods) with a pixel size equal to the SOFI pixel size. Scale bar, 1 μm. (**b**) Region of interest indicated in **a** showing a focal adhesion at different time points. Scale bar, 0.5 μm. (**c**) Kymographs along the direction of motion as indicated by the line in **b**. The focal adhesion mean velocity is determined by a linear fit to the centre of gravity *r* determined from the kymograph as a function of time (see Methods). The procedure was repeated five times for parallel kymographs, the error bar represents the s.d.

error have been developed for PALM, often based on merging localizations that are sufficiently close in time and space to be considered originating from the same blinking fluorescent protein[13,14]. As these methods require characterization of the blinking behaviour, for instance through the calculation of the average time between two emission bursts, they indirectly allow one to probe the molecular environment of the emitters.

Focal adhesions are dense assemblies of proteins, making it challenging to avoid merging localizations of different fluorescent proteins, which would lead to an under-counting error. Therefore, we have adapted the merging criterion of an earlier published work[13] to account for higher densities. Instead of using a fixed distance threshold of 1 raw image pixel as merging criterion, we assumed a threshold based on a statistical measure, called the Hellinger distance, which allows one to account for the varying localization precision (Supplementary Note 5). We applied this adapted method to our localization data (identical to those used for Fig. 1) of fixed MEF cells expressing paxillin labelled with mEos2 (Fig. 5a,b) and psCFP2 (Supplementary Fig. 7). The corrected localization number and the average time between two blinking events is shown as a function of different thresholds of the Hellinger distance, calculated for three areas with different emitter densities. We determined that a threshold value of 0.9 was a good compromise (Supplementary Note 5), but even around this value the number of localizations decreases with increasing threshold values for the densest areas (Fig. 5e). This indicates that the sample is too dense, which is corroborated by the average time between two blinking events being dependent on the area density (Fig. 5f).

SOFI is an interesting complement to PALM for quantitative imaging, since combining cumulant images of second, third and fourth order enables to extract molecular parameters such as the on-time ratio, the molecular brightness and the molecular density[21] (Supplementary Note 2). While PALM yields average values over the region of analysis, SOFI generates spatial maps of these parameters. Moreover, as SOFI is superior to PALM in imaging 'crowded' environments, this method is of great interest for quantitative imaging of focal adhesions. We used SOFI to determine the on-time ratio and density map of the same localization data used for PALM (Fig. 5c,d). As opposed to PALM, SOFI performs well in high-density areas. SOFI estimates the molecular parameters pixel-wise. This estimation is meaningless for areas that contain mostly background (SNR close to 1). Background areas therefore have to be removed by applying an intensity threshold or SNR based threshold. Since PALM is working well in these low-density areas, this again demonstrates the usefulness of our PALM-SOFI approach.

The molecular parameter estimation can be applied to living cells if the temporal resolution is sufficient for proper time sampling. Achieving fourth order SOFI images, required for molecular parameter estimation, is challenging in living cells since it requires high signals and generally a large number of frames. Given a high enough signal, 1,000 frames might be sufficient for the required fourth order. However, under our conditions in focal adhesions, 4,000–5,000 frames are necessary for high quality fourth order SOFI, which limits the temporal resolution of molecular parameter estimation. In the case of PALM, quantitative imaging requires a sufficient number of localizations, so the minimum number of frames will depend on the emitter density. On the other hand, if the density is too high, results will be biased. We therefore performed simulations to investigate SOFI and PALM-based molecular density estimation in function of the temporal resolution (that is, various numbers of frames) and the emitter density, see Fig. 5g and Supplementary Note 2. PALM-based density estimation performs well for low emitter densities (that is, $< 400 \, \mu m^{-2}$), regardless of the number of frames, while SOFI-based density estimation performs better than PALM for higher molecule densities, under the condition that enough frames are acquired (that is, $> 5{,}000$), as can be seen in Fig. 5g.

## Discussion

Our results indicate that PALM and SOFI are complementary techniques for the observation of focal adhesions in living cells. Such an imaging approach not only provides sufficient spatial resolution for their observation, it also grants access to their temporal dynamics. In view of the biological quest, we thoroughly investigated this imaging concept. Our simulations indicate a superior performance of SOFI when compared with PALM for low frame numbers (typically $< 5{,}000$ frames), while PALM substantially outperforms SOFI for higher frame numbers. The onset of 'super-resolution' based on SOFI demands typically 500 frames, while PALM requires at least 1000 frames. Our PALM-SOFI framework applied to the same raw image sequences therefore opens the door for assessing the dynamics of 'not too fast' biological processes in the order of 100 nm min$^{-1}$.

Using both PALM and SOFI, we could image focal adhesions with a resolution better than 100 nm in fixed cells, whereas in living cells a resolution $< 150$ nm was obtained, requiring $< 1000$ raw images. These live-cell images were recorded at a frame rate of 100 Hz, which translates into a temporal resolution below 10 s. Such a temporal resolution is required to resolve the dynamics of the focal adhesions in more detail, as we observed 'focal adhesion velocities' around 190 nm min$^{-1}$.

Besides resolution, we also characterized the SNR for our PALM-SOFI framework. In general, PALM shows the highest SNR, up to 25 dB for large frame numbers for fixed-cell images. Only for small frame numbers (typically $< 500$) SOFI showed a superior SNR. We attribute this difference to the different nature of PALM and SOFI images (that is rendered images based on localized emitters versus correlations of intensity fluctuations). Considering this difference, the ramp up towards the SNR plateau seems to be more important for our data than a comparison of absolute SNR values (Supplementary Fig. 8). The steeper onset of SNR is in favour of PALM whereas for SOFI the SNR plateau is reached at a lower frame number.

We used a generalized resolution metric named sFRC (adapted from the classical FRC metric) and a SNR metric based on statistical resampling for assessing the performance gain of the PALM-SOFI framework. Our simulations show that the sFRC metric is in agreement with the cutoff frequencies obtained from our MTF analysis (Supplementary Fig. 5). Under the tested conditions corresponding to focal adhesions, the (s)FRC values are slightly higher than expected. We attribute this to the fixed threshold used in the calculation of the (s)FRC metric. We would like to note that the sFRC by definition requires images with a rich spatial frequency content. When, for instance, a sparse structure in the presence of mostly background is imaged, the sFRC value is unreliable, and this metric is useful for qualitative comparison only.

Depending on the fluorophore properties, either PALM or SOFI yielded a better resolution. Using mEos2, PALM performed better, while SOFI outperformed PALM for psCFP2. We hypothesize that this is caused by a difference in activation rate, combined with a difference in fluorophore density. For mEos2, the localization number per frame was low and constant, which is in favour of PALM. psCFP2, on the other hand, showed a higher number of localizations during the beginning of the image acquisition, resulting in a better resolution for SOFI. This points to the interesting fact that difficult PALM data, caused by a 'crowded' environment, can still be evaluated by

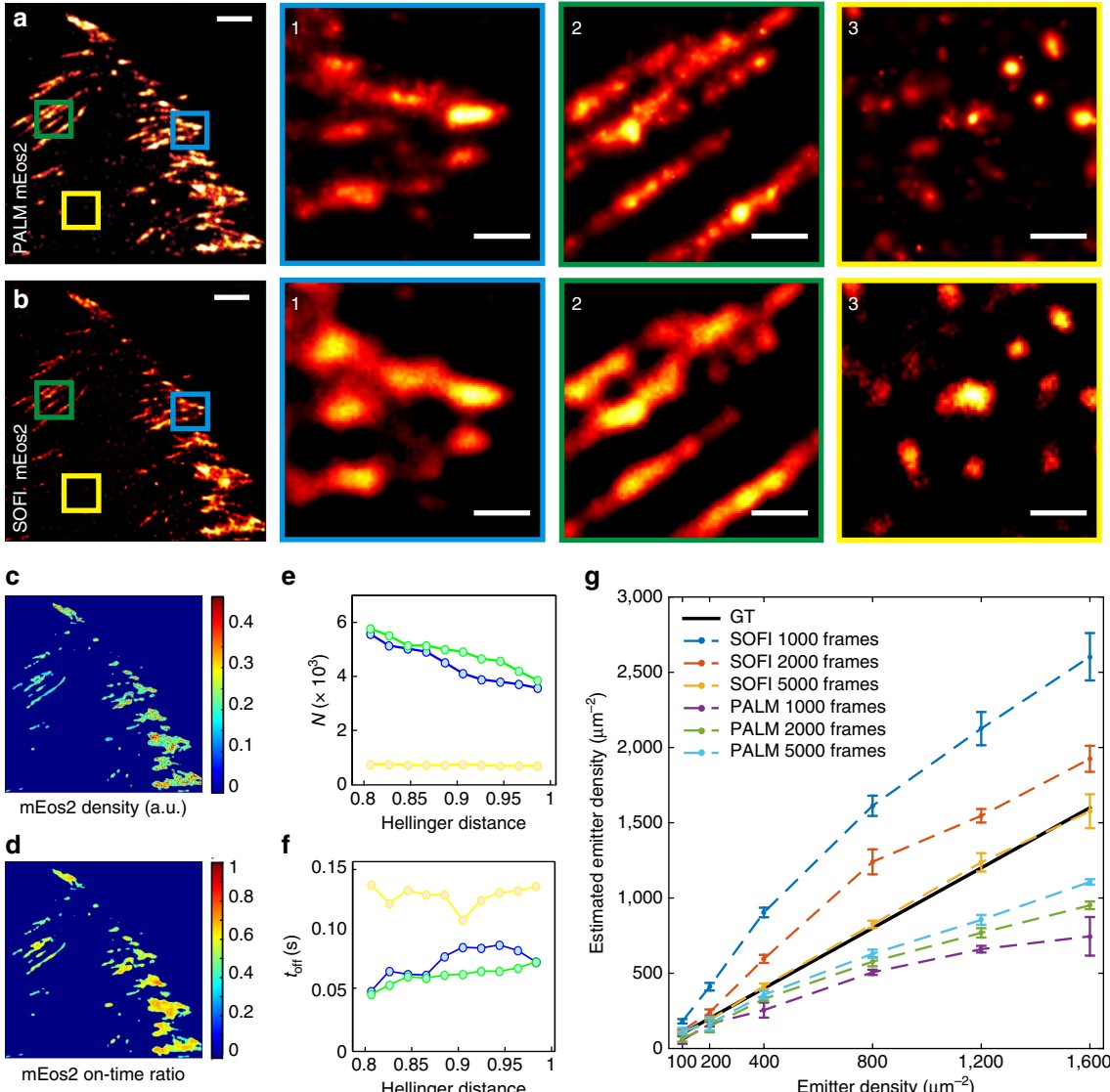

**Figure 5 | Quantitative imaging with PALM and SOFI.** (**a,b**) PALM and SOFI images of a fixed MEF cell expressing paxillin labelled with mEos2. Panels 1–3 are corresponding zoom-ins for the PALM or SOFI images. The PALM images were rendered as probability maps (see Methods). Scale bar, (**a,b**) 2 μm; (1–3) 0.5 μm. (**c,d**) SOFI analysis yields a fluorophore density and on-time ratio map. (**e,f**) Blinking events in PALM data can be detected by merging localizations that are sufficiently close in space and time. This analysis yields the blink corrected number of localizations $N$ and the corresponding average off-time $t_{off}$ between blinks, shown as a function of the Hellinger distance threshold for merging localizations. (**g**) SOFI and PALM-based quantitative analysis performed on simulated data (Supplementary Note 2). The emitter density estimated by PALM and SOFI is shown in function of the ground truth (GT) density for different numbers of simulated frames. The procedure was repeated 10 times, the error bars represent the s.d. The Hellinger distance threshold for the PALM-based estimation is 0.90 (Supplementary Note 5).

SOFI. In addition, our PALM-SOFI framework conveniently exploits these differences in fluorophore properties in order to improve on multicolour imaging, where one rarely has the luxury to choose an optimal combination of fluorescent proteins. As shown in Fig. 2, this allows to image both integrin β3 and paxillin in focal adhesions, without compromising the spatial resolution in one of the two-colour channels, which would be unavoidable when using only PALM or SOFI.

Another important benefit of this PALM-SOFI complementarity has been demonstrated by applying quantitative analysis on our focal adhesion data. PALM was shown to give reliable estimates of the blinking corrected localization numbers and the off-time between blinks in low-density areas of the cell sample. SOFI, on the other hand, was able to extract on-time ratios and number densities in high-density regions.

In summary, this PALM-SOFI imaging approach underlines the complementarity of both methods, enhanced by an additional gain in functional information. PALM imaging provides a high spatial resolution if enough frames can be acquired, while SOFI provides an interesting spatial resolution at lower frame numbers. The additional functional parameters provided by PALM and bSOFI post-processing add novel insights into cell biology without additional experimental effort.

## Methods

**Microscope set-up.** Fixed-cell imaging was carried out on a custom built microscope[34]. Three continuous wave laser sources were used for excitation/activation: a 50 mW 405 nm laser beam (Cube, Coherent), a 100 mW 488 nm laser beam (Sapphire, Coherent), a 100 mW 561 nm laser beam (Excelsior, Spectra Physics). The 488 and 561 nm laser beams were combined using a dichroic mirror (T495lpxr, Chroma) and sent through an acousto-optic tunable filter

(AOTFnC-VIS-TN, AA Opto Electronic). Both laser beams were combined with the 405 nm laser beam using a dichroic mirror (405 nm laser BrightLine, Semrock). The three laser beams were focused by a lens into the back focal plane of the objective mounted on an inverted optical microscope (IX71, Olympus). We used a $\times 100$ objective (UApo N $\times 100$, Olympus) with a numerical aperture of 1.49 configured for total internal reflection fluorescence microscopy. The fluorescence light collected by the objective was filtered to suppress the residual illumination light using a combination of a dichroic mirror (493/574 nm BrightLine, Semrock) and an emission filter (405/488/568 nm StopLine, Semrock). The fluorescence light was detected by an EMCCD camera (iXon DU-897, Andor). The back-projected pixel size was 105 nm. An adaptive optics system (Micao 3D-SR, Imagine Optics) and an optical system (DV2, Photometrics) equipped with a dichroic mirror (617/73 nm BrightLine, Semrock) were placed in front of the EMCCD camera. The Micao 3D-SR system was used to compensate for aberrations and the DV2 system was used to split the fluorescence light into a green and red-colour channel that were each sent to a separate half of the camera chip.

Live-cell imaging was carried out on a custom built microscope equipped with a temperature and $CO_2$ controlled incubator for live-cell imaging[20]. Three continuous wave laser sources were used for excitation/activation: a 120 mW 405 nm laser beam (iBeam smart, Toptica), a 200 mW 488 nm laser beam (iBeam smart, Toptica), a 800 mW 532 nm laser beam (MLL-FN-532, Roithner Lasertechnik). The 488 and 532 nm laser beams were combined using a dichroic mirror (T495LP, Chroma), and both laser beams were combined with the 405 nm laser beam using a dichroic mirror (T425LPXR, Chroma). All three laser beams were focused by a lens into the back focal plane of the objective. We used a $\times 60$ objective (Apo N $\times 60$, Olympus) with a numerical aperture of 1.49 configured for total internal reflection fluorescence illumination. The fluorescence light collected by the objective was filtered to suppress the residual illumination light using a combination of a dichroic mirror (Z488/532/633RPC, Chroma) and an emission filter (ZET405/488/532/640 m, Chroma). The fluorescence light was detected by an EMCCD camera (iXon DU-897, Andor). The back-projected pixel size was 96 nm.

**Sample preparation.** The MEF cells (kindly provided by Dr Luca Scorrano) and REFs (CRL-1213, ATCC) were grown in DMEM supplemented with 10% fetal bovine serum, 1% penicillin–streptomycin, 1% non-essential amino acids and 1% glutamine, at 37 °C with 5% $CO_2$. The cells were transfected by electroporation (Neon Transfection System, Invitrogen), which was performed on 600,000–800,000 cells using 1 pulse of 1,350 V lasting for 35 ms. The amount of DNA used for the transfection was 2 μg for the mEos2-paxillin-22 vector, 3 μg for the mEos2-Integrin-β3-N-18 vector, and 5 μg for the psCFP2-paxillin-22 vector.

For fixed-cell experiments, a 25 mm diameter microscope cover slip (#1.5 Micro Coverglass, Electron Microscopy Sciences) was prepared by first cleaning with an oxygen plasma for 5 min and then incubated with PBS containing 50 μg ml$^{-1}$ fibronectin (Bovine Plasma Fibronectin, Invitrogen) for 30 min at 37 °C. To remove the excess of fibronectin, the cover slip was washed with PBS. The transfected cells were seeded on the cover slip and grown in DMEM supplemented with 10% fetal bovine serum, 1% non-essential amino acids and 1% glutamine, at 37 °C with 5% $CO_2$. The cells were washed with PBS around 24 h after transfection, and then incubated in PBS with 4% paraformaldehyde at 37 °C for 30 min. After removing the fixative, the cells were again washed with PBS, and the cover slip was placed into a custom made holder.

For live-cell imaging, the transfected cells were seeded in a chambered cover slip system (Lab-Tek II Chambered Coverglass System, Thermo Scientific) and grown in DMEM supplemented with 10% fetal bovine serum, 1% non-essential amino acids and 1% glutamine, at 37 °C with 5% $CO_2$. Finally, the cells were washed with PBS around 24 h after transfection.

**Imaging procedure.** Fixed cells were imaged in PBS at room temperature. Before imaging, 100 nm gold nanospheres (C-AU-0.100, Corpuscular) had been added to the sample for lateral drift monitoring. Axial drift correction was ensured by a nanometre positioning stage (Nano-Drive, Mad City Labs) driven by an optical feedback system[34]. Excitation of mEos2 was done at 561 nm with ∼15 mW power (as measured in the back focal plane of the objective). Imaging of psCFP2 was performed using 488 nm laser light with ∼15 mW power. Both fluorescent labels were gently activated by 405 nm laser light with ∼5 μW power in case of single colour imaging, while ∼1.5 mW power was used for dual-colour imaging. The gain of the EMCCD camera was set at 100 and the exposure time to 50 ms. For each single colour experiment, at least 20,000 camera frames were recorded. Dual-colour imaging was performed similarly to a procedure published elsewhere[34]. First, at least 10,000 camera frames in the red channel were acquired in order to image mEos2, and subsequently the remaining population of mEos2 in the off-state was photobleached using 488 nm laser light. Finally, at least 10,000 camera frames were recorded in the green channel to image psCFP2. In addition, gold nanospheres visible in the red and green channel were imaged to co-register the two colour channels *a posteriori*.

The live cells were imaged in DMEM with low fluorescence background (FluoroBrite DMEM, Thermo Scientific) at 37 °C with 5% $CO_2$. Before imaging, 100 nm gold nanospheres (C-AU-0.100, Corpuscular) were added to the sample for lateral drift correction. mEos2 was excited at 532 nm with ∼8.5 mW power and activated by 405 nm laser light with ∼0.6 mW power. The gain of the EMCCD

camera was set at 150 and the exposure time to 10 ms. For each experiment at least 8,000 camera frames were recorded.

**PALM data analysis.** The recorded images were analysed by a custom written algorithm (Matlab, The Mathworks) that was adapted from an algorithm that was published elsewhere[5]. First, peaks were identified in each camera frame by filtering the frames and applying an intensity threshold. Only peaks with an intensity of at least 4 times the background were considered to be fluorescent labels or gold nanospheres. Subsequently, the peaks were fitted by maximum likelihood estimation of a 2D Gaussian point spread function (PSF)[35]. Drift was corrected in each frame by subtracting the average position of the gold nanospheres from the positions of the fluorescent labels that were localized in that frame. Co-registration of the two-colour channels was done using a second order polynomial transformation that was derived from the localizations of the gold nanospheres visible in both colour channels, using the Matlab function cp2tform. The theoretical localization precision for each fluorescent label was obtained from the Cramér-Rao lower bound of the maximum likelihood procedure[36]. This value was multiplied with the square root of 2 in order to account for the degradation of the localization precision caused by the electron multiplication process in the EMCCD camera[35]. The PALM images were generated either as a 2D localization number histogram or as a probability map by plotting a 2D Gaussian PSF centred on each fitted position with a standard deviation equal to the corresponding localization precision. Only positions with a localization precision between 1 and 50 nm were plotted.

**SOFI data analysis.** For the SOFI calculation, we adapted and enhanced the bSOFI algorithm[21] (Supplementary Note 2; Supplementary Figs 9–12). The whole input image sequence was divided into subsequences of 500 frames each. The subsequences were chosen sufficiently short to minimize the influence of photobleaching. As SOFI assumes the sample to be stationary over the investigated image subsequence, drift has to be corrected before the bSOFI processing. Tracking the positions of the gold nanospheres provides translational motion vectors in between consecutive frames. Drift was then corrected by registering the frames with sub-pixel precision using a bilinear interpolation. Co-registration of two-colour channels was done by applying the second order polynomial transform that was derived for PALM to the SOFI images. The linearization step of the bSOFI algorithm was replaced by an adaptive linearization, which takes into account blinking properties of the sample and thus enables more effective use of the available dynamic range and SNR for high-order SOFI analysis (Supplementary Fig. 9).

**Simulations.** For each fluorophore, a time trace was modelled, describing the number of photons emitted by a given fluorophore over time. The simulation assumed photokinetics typical for fluorescent proteins in PALM experiments (Supplementary Note 4; Supplementary Fig. 13). The intensity of a pixel at a certain time point was given by an integration of brightness from all fluorophores with a PSF that extends to that pixel at that time point. The number of photo-electrons was simulated by a Poisson distributed random number with an average value equal to the pixel value multiplied by the detection efficiency and added to the thermal noise (dark current). Gain noise and read-out noise were modelled as additive Gaussian noise. The parameters of the optical system and the camera used for simulations matched the properties of the microscope set-up. We tested two emitter densities: 800 and 1200 μm$^{-2}$, leading to two different scenarios (Fig. 3c,d). For each scenario, the number of photons per emitter per frame (that is, $I_{on}$) varied from 50 to 400 and the number of frames ranged from 500 to 20,000. In total, we generated and analysed 60 image stacks. Each image sequence was processed by a SMLM and a bSOFI algorithm. For SMLM processing, we used the FALCON algorithm[37] with the settings tuned for high densities. Using the bSOFI algorithm, images of second to sixth order were calculated. The cutoff frequency $f_c$ was measured for every bSOFI order. With increasing order of the bSOFI analysis, the resolution increases, but the image SNR decreases which limits the highest achievable resolution. The output SOFI cutoff frequencies shown in Fig. 3c,d represent the highest cutoff frequency achieved from the measured orders of the bSOFI analysis.

**Measuring the cutoff frequency.** An average line profile was calculated from each simulated super-resolved output image. The one-dimensional MTF (Supplementary Note 4) was calculated as the modulus of the discrete Fourier transform of the average line profile. Each MTF curve was smoothed by a moving average filter with a window length equal to 3 to suppress fluctuations and provide a more robust estimate of the cutoff frequency. To eliminate small non-zero MTF values that are caused mostly by noise and do not contain relevant information, we subtracted a constant 0.5 from each MTF curve before normalization. Each MTF curve was normalized using the MTF corresponding to the 20,000 frames test case as a reference. The cutoff frequency is the spatial frequency where the normalized MTF curve falls below a threshold (that is a small positive constant close to zero). The threshold was determined as the value of the widefield MTF, which occurs at the theoretical cutoff frequency of a noiseless diffraction-limited imaging system given by Abbe's resolution limit.

**sFRC calculation.** We used the sFRC metric for analysing the images shown in Fig. 1a,b. The full raw image sequence (20,000 frames) was split into 40 subsequences of 500 frames each (Supplementary Note 1; Supplementary Fig. 14). For bSOFI, images up to the sixth order were calculated for each subsequence. These images were split into two groups and averaged within each group to generate two SOFI images. The splitting procedure is described in Supplementary Note 1. For PALM, the localizations corresponding to the selected 500 frame subsequences were pooled and used to render two independent PALM images as 2D histograms with a pixel size that is ∼1/6 of the real pixel size, matching the sixth order bSOFI pixel size. To minimize the effect of photobleaching during the image sequence, the recombination into two independent PALM/SOFI images was done in an alternating way, and an extra correction was applied in case of SOFI, see Supplementary Note 1. To observe the evolution of the sFRC with increasing number of frames, the calculation was repeated using an increasing amount of frames, going from 1,000 to 20,000 frames with an increment to 1,000 frames in each step. The sFRC was calculated in separate sectors with an angular extent of $\pi/12$. The results for all sectors are shown in Supplementary Fig. 3. Two selected sectors are shown in Fig. 1c.

**SNR calculation.** We calculated the pixel-wise SNR using a statistical approach, that is, jackknife resampling (Supplementary Note 3; Supplementary Fig. 15) on the data shown in Fig. 1a,b. For an objective comparison, PALM images were rendered as histograms with a pixel size of 105 nm (that is, the pixel size in the raw images) and SOFI images were binned on an equal pixel size before the SNR estimation. To observe the evolution of the SNR throughout the raw image sequence (20,000 frames), the calculation was repeated for an increasing number of frames, starting with 1,000 frames and adding the next 1,000 frames in each step. The SNR values as function of the number of frames are shown in Fig. 1d.

**Kymograph-based analysis.** The kymograph shown in Fig. 4c along the line indicated in Fig. 4b was obtained using ImageJ (National Institutes of Health). For each time point, the centre position of the focal adhesion was calculated as the centre of gravity $r$ along the corresponding line in the kymograph, with the PALM/SOFI pixel values as weights. The focal adhesion mean velocity was determined as the slope of a linear fit to these centre positions, as a function of the time points. This procedure was repeated for four other lines parallel to the one shown in Fig. 4b. The reported focal adhesion mean velocity is the average, and the error bar represents the corresponding s.d. The direction of the kymograph was chosen as the direction of the focal adhesion mean velocity, which was determined by applying the above procedure to the x- and y-direction separately (Supplementary Fig. 16).

**Data availability.** All data and code are available from the corresponding author upon request.

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

## Acknowledgements

The MEF cells were kindly provided by Dr Luca Scorrano. The mEos2-paxillin-22, mEos2-Integrin-β3-N-18, and psCFP2-paxillin-22 vectors were kindly provided by Dr Michael Davidson. H.D. and A.R. acknowledge the support of the Max Planck-EPFL Center for Molecular Nanoscience and Technology. A.R. and T.La. acknowledge the support of the Horizon 2020 project AD Gut (SEFRI 16.0047). We highly appreciate the partial funding by the Swiss National Science Foundation (SNSF, http://www.snf.ch/) under grants 200020-159945 and 205321-138305. T.Lu. acknowledges a SCIEX scholarship (13.183) and a Czech Technical University student grant (SGS16/167/OHK3/2T/13).

## Author contributions

H.D., T.Lu., T.La. and A.R. conceived the study. T.Lu., T.La. and D.S. developed the MTF analysis. T.Lu. performed the simulations. H.D. and L.F. prepared the samples.

H.D. performed the fixed-cell experiments, H.D. and A.S. performed the live-cell experiments. T.Lu. developed the enhanced bSOFI algorithm. T.Lu. and H.D. analysed the data. W.V., M.L. developed the Jackknife code. D.S., W.V., P.D., J.H. and M.L. provided research advice. H.D., T.Lu., T.La. and A.R. wrote the paper. All authors reviewed and approved the manuscript.

## Additional information

**Competing financial interests**: The authors declare no competing financial interests.

**How to cite this article**: Deschout, H. *et al.* Complementarity of PALM and SOFI for super-resolution live-cell imaging of focal adhesions. *Nat. Commun.* **7,** 13693 doi: 10.1038/ncomms13693 (2016).

**Publisher's note**: 

