## [Peer Review File · Nature Communications]

Reviewers' comments:

Reviewer #1 (Remarks to the Author):

The paper from Deschout et al. compares the performance of two super-resolution microscopy methods, namely SOFI and PALM, under different imaging conditions. The main goal is to assess in which respect they are complementary to each other, and to elucidate the strengths and weaknesses of both methods. The evaluation is carried out by using simulations as well as imaging of fixed and living cells, in particular the focal adhesions complex in these cells. It is shown that SOFI is more suited for high density labeling and short movies, i.e. low frame numbers, while PALM is the superior method for low density labeling and when large numbers of frames are available.

The paper is well written and easy to understand. The evaluation is exhaustive and detailed. There exists, until now, only one publication which is concerned with a quantitative and thorough comparison of the performance of both super-resolution microscopy methods ("Comparison between SOFI and STORM", Geissbuehler et al., Biomed Opt Express. 2011 Mar 1; 2(3): 408-420), but this publication was focused on completely different experimental parameters and was mostly theoretical. The strength of the present manuscript is the comparison of both methods within the context of real-life cell imaging, and by focusing on a concrete biological structure, the focal adhesion complex. Thus, the paper will help researchers to select the most suitable method for the imaging conditions and particular problems they are interested in.

However, a few questions remain which should be addressed by the authors:

- It should be stated how the PALM images were rendered. Was this done by simple histogramming, or was a more complex method used?
- For the quantitative imaging in Part 2.4: How many frames were used to estimate the molecular parameters? Can this be done with the 1000 frames used to evaluate the movement of the focal adhesion, or is this only possible for fixed cells? Given the quality of the bSOFI2 images in Figure 1(b), and that fourth order SOFI is necessary for the estimation, I would assume it is not possible with this temporal resolution. Is this correct? Please state this clearly in the paper.
- The bSOFI2 image in Figure 1(a) looks like it is mostly showing artifacts of deconvolution. Does the non-deconvolved image look similar? It would be good to show both images for comparison.
- The manuscript part which shows the molecular density and the on-time ratio maps can be improved by presenting a comparison of the performance of both methods using simulations, similar to what was done in part 2.2 for resolution.

Provided that these questions are appropriately answered, I recommend publication of the manuscript.

Reviewer #2 (Remarks to the Author):

Summary: Live cell superresolution microscopy using genetically encoded photoswitchable fluorescent proteins are commonly performed by single-molecule localization analysis (PALM), or, in several prior studies, SOFI, which relies on the analysis of the stochastic blinking of the fluorophores. In Deschout et al, the authors established how both analysis approaches can be applied to the same dataset of live cells expressing photoswitchable fluorescent proteins. The authors quantified spatial resolution and signal-to-noise ratio of both approaches using rigorous and objective criteria that also take into

account the blinking characteristics of the fluorophores. Live-cell superresolution imaging is carried out on fibroblast cells expressing paxillin tagged with either mEos2 or PSCFP2 fluorescent proteins, revealing translocation of focal adhesion at a rate of ~ 190 nm/ min.

Reviewer's comments:

Although this study represents a careful methodology development that should be of great interest to the superresolution microscopy community, there are a number of aspects that, in this reviewer's opinion, make this manuscript more appropriate for a specialized audience and not quite ready for Nature Communications in its current form.

Foremost, although the methods are applied to image the focal adhesions, it doesn't appear to have been applied with a goal toward uncovering any new insights into the biological functions/mechanisms of focal adhesions, nor have the authors suggest in specific terms how one might go about doing that. In particular, the authors have limited themselves to focus on just one protein, paxillin.

Furthermore, although focal adhesions are dynamic structures, their minute-scale dynamics are relatively slow compared to many biological processes that have since been studied by live superresolution techniques such as endocytosis. Indeed, in no small part due to its relatively slow dynamics, the focal adhesions have been among the first structures imaged live by superresolution methods in the late 2000s (Betzig lab, Shroff et al., Nature Methods 2008), where they have looked at paxillin, observing focal adhesions translocation (e.g. in the supplementary videos therein) that is not too dissimilar from the results reported in this study.

Given that recent development in superresolution microscopy such as SIM has achieved much faster speed, and nearly comparable spatial resolution, this reviewer feels that the authors should need to make a stronger case as to the advantages of the combined PALM/SOFI method presented here. For example, there are many key proteins in focal adhesions beyond paxillin, and if the PALM/SOFI method can be applied to systematically look at their comparative dynamics (or better yet with dual-channel capability) to reveal new insights or to quantify important properties that are difficult to obtain otherwise, this would help make a more compelling case for this approach.

Response to the reviewers' comments

We would like to thank the reviewers for carefully reading our manuscript and for their constructive comments which motivated us to perform additional experiments and simulations generating novel insights.

We would like to emphasize the two main changes of our revised manuscript:

1. We launched new simulations in order to evaluate and compare the reliability of the SOFI based molecular parameter estimation and the PALM based molecule counting analysis. The results demonstrate the advantage of our complementary approach for quantitative imaging. PALM based density estimation performs well for low molecular density areas, regardless of the number of frames, while SOFI based density estimation performs better than PALM for high molecular densities and high frame numbers.
2. We performed and added dual-color experiments, simultaneously imaging paxillin and integrin, essential proteins in the formation and functioning of focal adhesions. Even more important, these dual-color experiments allowed us to identify a new feature of our complementary SOFI/PALM framework. Conventional dual-color PALM is restricted in the choice of fluorescent proteins, which often reduces spatial resolution in at least one color channel. Our complementary SOFI/PALM framework overcomes this problem, since sub-optimal fluorescent proteins for PALM may result in high quality SOFI images, leading to an improved dual-color quality.

We believe that these new results underline the advantages of our SOFI/PALM framework. Detailed answers to all reviewers' comments follow. All changes introduced to the manuscript have been highlighted in yellow.

Reviewer 1:

The paper is well written and easy to understand. The evaluation is exhaustive and detailed. There exists, until now, only one publication which is concerned with a quantitative and thorough comparison of the performance of both super-resolution microscopy methods ("Comparison between SOFI and STORM", Geissbuehler et al., Biomed Opt Express. 2011 Mar 1; 2(3): 408-420), but this publication was focused on completely different experimental parameters and was mostly theoretical. The strength of the present manuscript is the comparison of both methods within the context of real-life cell imaging, and by focusing on a concrete biological structure, the focal adhesion complex. Thus, the paper will help researchers to select the most suitable method for the imaging conditions and particular problems they are interested in.

We truly appreciate this positive evaluation of our work.

However, a few questions remain which should be addressed by the authors:

- It should be stated how the PALM images were rendered. Was this done by simple histogramming, or was a more complex method used?

We thank the reviewer for pointing to this issue. The PALM images in the original **Figures 1 and 4** were rendered as probability maps by plotting a 2D Gaussian PSF centered on each fitted position with a standard deviation equal to the corresponding localization precision. The PALM images in the original **Figures 2 and 3** were rendered as 2D localization number histograms with a bin size equal to the pixel size in the SOFI images, in order to allow a straightforward comparison. We have added this information to the captions of the revised figures as follows:

*"The PALM images were rendered as probability maps (see **Methods**)."*

or:

*"The PALM images were rendered as localization number histograms (see **Methods**) with a pixel size equal to the SOFI pixel size."*

For the quantitative imaging in Part 2.4: How many frames were used to estimate the molecular parameters? Can this be done with the 1000 frames used to evaluate the movement of the focal adhesion, or is this only possible for fixed cells? Given the quality of the bSOFI2 images in Figure 1(b), and that fourth order SOFI is necessary for the estimation, I would assume it is not possible with this temporal resolution. Is this correct? Please state this clearly in the paper.

Yes, this is correct. For the molecular parameter estimation shown in **Section 2.4**, we used 10,000 frames. Fourth order SOFI is indeed necessary for this estimation. Molecular estimation is applicable to live cells if the temporal resolution is sufficient for proper sampling of temporal changes of the sample in question. Given a strong enough signal, 1000 frames might be sufficient, but under the current conditions for focal adhesions 4000 – 5000 frames are usually needed for high quality fourth order SOFI,

which limits temporal resolution of the analysis of molecular parameters. As requested by the reviewer, we added the following statement to **Section 2.4** of the revised manuscript:

“The molecular parameter estimation can be applied to living cells if the temporal resolution is sufficient for proper time sampling. Achieving 4th order SOFI images, required for molecular parameter estimation, is challenging in living cells since it requires high signals and generally a large number of frames. Given a high enough signal, 1000 frames might be sufficient for the required fourth order. However, under our conditions in focal adhesions, 4000-5000 frames are necessary for high quality fourth order SOFI, which limits the temporal resolution of molecular parameter estimation. In the case of PALM, quantitative imaging requires a sufficient number of localizations, so the minimum number of frames will depend on the emitter density. On the other hand, if the density is too high, results will be biased.”

Additionally, as recommended by the reviewer in the comment below, we investigated the molecular density estimation in function of the temporal resolution, i.e. a various number of frames. The results are added to **Figure 5** of the revised manuscript.

The bSOFI2 image in Figure 1(a) looks like it is mostly showing artifacts of deconvolution. Does the non-deconvolved image look similar? It would be good to show both images for comparison.

For 1000 frames, the SOFI reconstruction in **Figure 1b** indeed shows some deconvolution artifacts, because the SNR for such a small number of frames was quite low, which is a challenge for any deconvolution task. The deconvolution artifacts are thus an indication that more frames are necessary. However, depending on the fluorophore properties, we found that SOFI can still outperform PALM for this number of frames (as in the case of psCFP2 in **Figure 1b**). We thank the reviewer for this remark. We have carefully reworked **Figure 1b** to ensure proper visualization of the result and we added a new **Supplementary Figure 15** which shows the comparison between the non-deconvolved raw SOFI image and the bSOFI image:

Supplementary Figure 15: (a) Raw SOFI image (2nd order) reconstructed from first 1000 frames of the input image sequence. (b) bSOFI image i.e. raw SOFI image in (a) deconvolved and linearized using the procedure described in **Supplementary Note 2.2**.

We also added the following accompanying remarks to the revised **Supplementary Note 2.2**:

*“The bSOFI algorithm contains an inherent deconvolution step [4]. Raw SOFI images were deconvolved using the MATLAB function “deconvlucy” and linearized according to the linearization procedure described above. The number of iterations for the deconvolution was set to 10 (standard settings). The PSF was modeled by a 2D Gaussian function and the FWHM was estimated from the data using the procedure described in [4]. **Supplementary Figure 6** shows a comparison of a raw SOFI image and a bSOFI image if 1000 frames are used for the reconstruction. For low number of frames and low signals, the low SNR areas may exhibit some deconvolution artifacts, which is usually an indication that more frames are required for the reconstruction. More advanced deconvolution with regularization can be considered in the future for further improvement of the results.”*

The manuscript part which shows the molecular density and the on-time ratio maps can be improved by presenting a comparison of the performance of both methods using simulations, similar to what was done in part 2.2 for resolution.

We fully agree that an evaluation of the reliability/precision of the molecular parameter estimation would be a valuable information, hence we performed the extra simulations as requested by the reviewer. On-time ratio estimation using PALM might be doable in principle, however its investigation is beyond the scope of the current manuscript. Moreover, this parameter is easily extracted from SOFI, a strong feature demonstrating the complementarity of both methods. Therefore, we focused on

estimating densities using both SOFI and PALM. The results are shown in panel “g” of the revised **Figure 5** in the manuscript:

Figure 5g: SOFI and PALM based quantitative analysis performed on simulated data (see **Supplementary Note 2**). The emitter density estimated by PALM and SOFI is shown in function of the ground truth (GT) density for different numbers of simulated frames. The Hellinger distance threshold for the PALM based estimation is 0.90 (see **Supplementary Note 5**).

Details on the simulations can be found in the revised **Supplementary Note 2.4**, and we also added the following discussion to **Section 2.4** of the revised manuscript:

*“We therefore performed simulations in order to investigate SOFI and PALM based molecular density estimation in function of the temporal resolution (i.e. various number of frames) and the emitter density, see **Figure 5g** and **Supplementary Note 2**. PALM based density estimation performs well for low molecular densities (i.e. < 400 #/μm²), regardless of the number of frames, while SOFI based density estimation performs better than PALM for higher molecule densities, under the condition that enough frames are acquired (i.e. > 5000), as can be seen in **Figure 5g**.”*

Reviewer 2:

Foremost, although the methods are applied to image the focal adhesions, it doesn't appear to have been applied with a goal toward uncovering any new insights into the biological functions/mechanisms of focal adhesions, nor have the authors suggest in specific terms how one might go about doing that. In particular, the authors have limited themselves to focus on just one protein, paxillin.

We are thankful for the remarks of the reviewer, and hope to clarify better in the following the advantages of our PALM/SOFI framework. Indeed, we intended our work to be a thorough methodological investigation on how the combination of PALM and SOFI can be used to image focal adhesions, without focusing on a specific function or mechanism. We chose to investigate paxillin, since this protein has often been imaged in former reports on PALM. In this way, we applied our PALM-SOFI framework to a known structure, making an objective assessment of the performance of both imaging techniques more obvious. However, we agree with the reviewer that imaging other proteins is likely to contribute to new insight into the biology of focal adhesions. We have therefore added new data on integrin $\beta 3$, in order to respond to these requests and to enforce our statements. This novel work has been added as a new **Figure 2**:

Figure 2: Dual-color imaging with PALM and SOFI. (a-c) Overlay of red and green images of a fixed REF expressing paxillin labeled with psCFP2 (green) and integrin $\beta 3$ labeled with mEos2

(red) as obtained by (a) PALM in both color channels, (b) SOFI in both color channels, and (c) PALM in the red channel and SOFI in the green channel. (d-f) Corresponding zoom-ins for the images in (a-c). The PALM images were rendered as probability maps (see **Methods**).

An accompanying discussion in **Section 2.1** was added as well:

*“Following these observations, we extended our PALM-SOFI framework to dual-color imaging using both psCFP2 and mEos2. We imaged a fixed rat embryonic fibroblast (REF) expressing paxillin labelled with psCFP2 and integrin β 3 labelled with mEos2 (see **Methods**). Calculating the sFRC metric in the two color channels for both SOFI and PALM shows that SOFI obtains the highest spatial resolution in the psCFP2 channel (i.e. around 90 nm), while PALM gives the best resolution (i.e. around 100 nm) in the mEos2 channel, in correspondence to our single color results above. This suggests that an overlay of the SOFI (psCFP2) and PALM (mEos2) images results in an improved dual-color image, as shown in **Figure 2.**”*

Furthermore, although focal adhesions are dynamic structures, their minute-scale dynamics are relatively slow compared to many biological processes that have since been studied by live superresolution techniques such as endocytosis. Indeed, in no small part due to its relatively slow dynamics, the focal adhesions have been among the first structures imaged live by superresolution methods in the late 2000s (Betzig lab, Shroff et al., Nature Methods 2008), where they have looked at paxillin, observing focal adhesions translocation (e.g. in the supplementary videos therein) that is not too dissimilar from the results reported in this study.

SMLM trades temporal for spatial resolution. Using our complementary PALM-SOFI framework to image focal adhesions, we achieved a temporal resolution of 10 s while maintaining spatial super-resolution. To our knowledge, this temporal resolution is higher than what was previously reported on the use of PALM for imaging focal adhesions. As mentioned by the reviewer, other structures have indeed been imaged at higher temporal resolutions using SMLM, most often (d)STORM. However, the delivery of (d)STORM dyes to intracellular targets in living cells such as focal adhesions is challenging. That is the reason why we opted for PALM, since it uses genetically expressed fluorescent proteins that are well tolerated by living cells.

Given that recent development in superresolution microscopy such as SIM has achieved much faster speed, and nearly comparable spatial resolution, this reviewer feels that the authors should need to make a stronger case as to the advantages of the combined PALM/SOFI method presented here. For example, there are many key proteins in focal adhesions beyond paxillin, and if the PALM/SOFI method can be applied to systematically look at their comparative dynamics (or better yet with dual-channel capability) to reveal new insights or to quantify important properties that are difficult to obtain otherwise, this would help make a more compelling case for this approach.

We thank the reviewer for pointing out this important issue. The key message is that our complementary PALM-SOFI approach (1) offers a high temporal resolution, while maintaining a high spatial resolution, and (2) allows quantitative imaging, which has not been reported for other super-

resolution microscopy techniques like SIM. We have added this extra statement on the advantages of our approach to **Section 3** of the revised manuscript:

“In summary, this PALM-SOFI imaging approach underlines the complementarity of both methods, enhanced by an additional gain in functional information. PALM imaging provides a high spatial resolution if enough frames can be acquired, while SOFI provides an interesting spatial resolution at lower frame numbers. The additional functional parameters provided by PALM and bSOFI post-processing add novel insights into cell biology without additional experimental effort.”

Furthermore, we have followed in detail the reviewer’s excellent suggestion and performed dual-color PALM-SOFI imaging. Interestingly, we found that our approach can improve the image quality in each color channel independently, by taking into account the strengths of SOFI and PALM. This is underlined by the overlay image combining PALM (integrin $\beta 3$) and SOFI (paxillin), clearly showing the gain in quality obtained by our complementary PALM-SOFI framework. We highlighted this result as a new **Figure 2** (as shown above). We also added the following text to **Section 3** of the revised manuscript:

*“Additionally, our PALM-SOFI framework conveniently exploits these differences in fluorophore properties in order to improve on multi-color imaging, where one rarely has the luxury to choose an optimal combination of fluorescent proteins. As shown in **Figure 2**, this allows to image both integrin $\beta 3$ and paxillin in focal adhesions, without compromising the spatial resolution in one of the two color channels, which would be unavoidable when using only PALM or SOFI.”*

REVIEWERS' COMMENTS:

Reviewer #1 (Remarks to the Author):

The authors have addressed and appropriately answered all my questions and comments. The addition of new data, in particular the addition of dual color imaging data, makes the manuscript now even significantly stronger. I recommend publication as is.

Reviewer #2 (Remarks to the Author):

In the revised manuscript, the authors have attempted to address the concern raised by this reviewer, primarily by performing 2-color experiments and conducted their analysis by PALM and SOFI. As this reviewer has stated since the original review, while the technical aspects of this manuscript has been solidly performed and well described, the demonstrated applications of the methods to gain novel biological insights are limited, and as such, in this reviewer's opinion this potentially limit the broader appeal of this study to the general readership of Nature Communications.

While the additional 2-color experiment is a step in the direction of demonstrating the utility of this approach, this reviewer still feels that more could have been done, given the premise of this study is to improve the temporal resolution, and to provide additional novel insights into focal adhesions, as stated by the authors. For example, while the additional experiments are performed in 2-color, this is performed in fixed cell, and thus no temporal information is available. While the need for fixation may be due to the requirement for sequential imaging of PSCFP2 and mEos2 fluorescent proteins, there are alternative pairs such as PA-GFP and PAmCherry which should allow live 2-color imaging.

Furthermore, many of the photoactivatable fluorescent protein constructs of focal adhesion proteins are publicly available in public depository such as AddGene. Alternatively, the authors could have demonstrated how the dynamics of focal adhesion proteins may vary as a function of some biological perturbations (drug treatment to affect cell contractility, actin dynamics, and so on), by comparing the treated vs control samples. Along a similar vein, the authors could have characterized the dynamics of focal adhesions as a function of their spatial position within the cells. For example, do all focal adhesions move at the same speed of 190 nm/min? Is the speed correlated to the position on adhesions near the leading edge or retracting rear of the cells? These are examples of the types of applications that this reviewer feel that the authors should demonstrate to support their claim on the utility of their new approach.

Response to the reviewers' comments

We would like to thank the reviewers for carefully reading our manuscript a second time and for their constructive comments. Answers to the reviewers' comments follow.

Reviewer 1:

The authors have addressed and appropriately answered all my questions and comments. The addition of new data, in particular the addition of dual color imaging data, makes the manuscript now even significantly stronger. I recommend publication as is.

We appreciate the positive evaluation of our revised manuscript.

Reviewer 2:

In the revised manuscript, the authors have attempted to address the concern raised by this reviewer, primarily by performing 2-color experiments and conducted their analysis by PALM and SOFI. As this reviewer has stated since the original review, while the technical aspects of this manuscript has been solidly performed and well described, the demonstrated applications of the methods to gain novel biological insights are limited, and as such, in this reviewer's opinion this potentially limit the broader appeal of this study to the general readership of Nature Communications.

While the additional 2-color experiment is a step in the direction of demonstrating the utility of this approach, this reviewer still feels that more could have been done, given the premise of this study is to improve the temporal resolution, and to provide additional novel insights into focal adhesions, as stated by the authors. For example, while the additional experiments are performed in 2-color, this is performed in fixed cell, and thus no temporal information is available. While the need for fixation may be due to the requirement for sequential imaging of PSCFP2 and mEos2 fluorescent proteins, there are alternative pairs such as PA-GFP and PAmCherry which should allow live 2-color imaging. Furthermore, many of the photoactivatable fluorescent protein constructs of focal adhesion proteins are publicly available in public depository such as AddGene. Alternatively, the authors could have demonstrated how the dynamics of focal adhesion proteins may vary as a function of some biological perturbations (drug treatment to affect cell contractility, actin dynamics, and so on), by comparing the treated vs control samples. Along a similar vein, the authors could have characterized the dynamics of focal adhesions as a function of their spatial position within the cells. For example, do all focal adhesions move at the same speed of 190 nm/min? Is the speed correlated to the position on adhesions near the leading edge or retracting rear of the cells? These are examples of the types of applications that this reviewer feel that the authors should demonstrate to support their claim on the utility of their new approach.

We are thankful for these comments which represent good suggestions for future research projects.